# Diet, Physical Activity and Adiposity as Determinants of Circulating Amino Acid Levels in a Multiethnic Asian Population

**DOI:** 10.3390/nu12092603

**Published:** 2020-08-27

**Authors:** Samuel H. Gunther, Chin Meng Khoo, Xueling Sim, E Shyong Tai, Rob M. van Dam

**Affiliations:** 1Saw Swee Hock School of Public Health, National University of Singapore, Singapore 117549, Singapore; ephsx@nus.edu.sg (X.S.); e_shyong_tai@nuhs.edu.sg (ES.T.); rob.van.dam@nus.edu.sg (R.M.v.D.); 2Department of Medicine, Yong Loo Lin School of Medicine, National University Health System, Singapore 119228, Singapore; Chin_Meng_KHOO@nuhs.edu.sg; 3Department of Nutrition, Harvard T.H. Chan School of Public Health, Harvard University, Boston, MA 02115, USA

**Keywords:** branched-chain amino acids, dietary protein source and intake, metabolomics

## Abstract

Profiles of circulating amino acids have been associated with cardiometabolic diseases. We investigated the associations between dietary protein intake, physical activity and adiposity and serum amino acid profiles in an Asian population. We used data from 3009 male and female participants from the Singapore Prospective Study Program cohort. Dietary and physical activity data were obtained from validated questionnaires; anthropometric measurements were collected during a health examination; and fasting concentrations of 16 amino acids were measured using targeted LC-MS. The association between lifestyle factors and amino acid levels was modeled using multiple linear regression with adjustment for other sociodemographic and lifestyle factors and correction for multiple testing. We observed significant associations between seafood intake (*β*-coefficient 0.132, 95% CI 0.006, 0.257 for a 100% increment), physical activity (*β*-coefficient −0.096, 95% CI −0.183, −0.008 in the highest versus lowest quartile) and adiposity (BMI *β*-coefficient 0.062, 95% CI 0.054, 0.070 per kg/m^2^; waist circumference *β*-coefficient 0.034, 95% CI 0.031, 0.037 per cm) and branched-chain amino acid levels (expressed per-SD). We also observed significant interactions with sex for the association between meat and seafood and total intakes and BCAA levels (*P* for interaction 0.007), which were stronger in females than in males. Our findings suggest novel associations between modifiable lifestyle factors and amino acid levels in Asian populations.

## 1. Introduction

Metabolic and cardiovascular diseases present urgent and growing public health challenges. The prevalence of diabetes mellitus is projected to reach 700 million globally by 2045 [1], and cardiovascular diseases such as coronary artery disease and stroke are projected to cause 23.6 million annual deaths by 2030 [2]. Modifiable lifestyle factors and behaviors have been recognized to contribute to metabolic and cardiovascular disease risk [3,4,5,6,7,8,9]. Obesity, for example, is a precursor of the metabolic syndrome and increases the risk of type 2 diabetes, heart disease, stroke and cancer [5,6], while physical inactivity and sedentary behavior increase diabetes risk independent of obesity [5]. Certain dietary intake patterns, such as consumption of red meat, have also been reported to be associated with increased risk of coronary heart disease, heart failure and stroke, as well as diabetes [7]. It has been suggested that the mechanisms which link these exposures to metabolic disease outcomes are mediated by insulin resistance [8,9]. Insulin-stimulated glucose uptake by skeletal muscle and adipocytes becomes diminished in conditions of obesity [8], while physical exercise has beneficial effects on insulin sensitivity and glycemic control [9]. However, much about the biological links between these and other lifestyle factors, such as dietary protein intake, and cardiometabolic disease risk remains to be understood [10].

One promising area of research to gain insight into these mechanisms involves applying metabolomics, the measurement and identification of circulating biomolecules, to observational and intervention studies of lifestyle exposures. Previous metabolomics studies have reported associations between lifestyle behaviors, particularly dietary intakes, and concentrations of the branched-chain amino acids (BCAAs) isoleucine, leucine and valine [11,12,13,14,15,16,17,18]. While these metabolites are innately involved in biological functions, BCAAs have also been consistently identified as risk factors for cardiometabolic diseases due to their strong association with insulin resistance [19], with accumulating evidence for causal effects [20,21,22,23]. Higher consumption of red meat has consistently been linked to higher BCAA levels [14,15,16,17]. However, these studies have mainly been conducted in populations where red meat is the largest source of protein. Moreover, amino acids other than the branched-chain species, including alanine, phenylalanine and tyrosine, have also been identified as potential cardiometabolic disease risk factors [13,24], but potential lifestyle determinants of these amino acids have rarely been examined. Furthermore, other modifiable factors such as physical activity and body fatness remain understudied in relation to amino acid levels [25]. The aim of the current study was to address these gaps in evidence by investigating dietary protein and its food sources, physical activity and adiposity in relation to serum concentrations of 16 amino acids in a multiethnic Asian population.

## 2. Materials and Methods

### 2.1. Study Population and Design

The study population consisted of participants from the Singapore Prospective Study Program (SP2), a population-based study conducted in Singapore between 2004 and 2007, the methodology of which has been previously described in detail [26]. Participants were contacted at their homes for an interview, which consisted of standardized questionnaires on sociodemographic and lifestyle factors and medical history. Subsequently, they were invited for a health examination, which included physical measurements and collection of blood samples. Fasting concentrations of serum amino acids were measured in a representative subset (n = 4351) of the 5157 SP2 participants who completed both the interview and the health examination. Of the 4351 participants with amino acid measurements, 125 were determined to have provided invalid dietary intake responses, indicating either non-random skip patterns or extreme energy intake (less than 500 or more than 5156.74 kilocalories per day of total energy for males and less than 500 or more than 4403.78 kilocalories per day for females, the upper cutoffs representing three SDs above the gender-specific mean energy intake) and therefore excluded. We further excluded 1217 participants diagnosed with medical conditions that may have altered their food consumption habits, including heart attack, stroke, type 2 diabetes mellitus and cancer, leaving 3099 participants available for this study. Figure 1 presents a flow chart describing the sample selection process. All participants provided written informed consent before taking part in SP2 and the follow-up examinations. SP2 and the follow-up examinations were approved by the National University of Singapore Institutional Review Board (NUS-IRB 12−282), and all studies were conducted in agreement with the Declaration of Helsinki conventions for the ethical conduct of medical research.

### 2.2. Assessment of Dietary Intakes

Participants’ dietary intake was assessed using a semi-quantitative food frequency questionnaire (FFQ), developed using weighted food records collected during the Singapore National Health Survey 1993 [27]. Reported food items were classified into various food groups based on ingredients and nutritional profiles. These food items were subsequently included on the food list if they represented important contributions to the overall intakes of total energy, total fat, saturated fat, polyunsaturated fat, monounsaturated fat and cholesterol, based on Ministry of Health, Singapore food composition data [28]. Foods contributing cumulatively to 90% of total intake of energy or one of the other five nutrients were included on the list [27].

The result was a 169-item semi-quantitative FFQ that was validated against 24-h dietary recalls in a random sample of 457 adults from the general Singapore population. Validity was determined by comparing the intakes based on the FFQ against the responses for the 24-h recalls. Reasonable correlations were found for total energy (*r* = 0.56) and key nutrients (total fat: *r* = 0.58; saturated fat: *r* = 0.51; polyunsaturated fat: *r* = 0.39; monounsaturated fat: *r* = 0.50; cholesterol: *r* = 0.51; protein: *r* = 0.46) [27]. Food and nutrient intakes and total energy intake for each study participant were derived from portion sizes and frequency of consumption and standardized to grams per day. We calculated protein intake as a percentage of total energy. In addition, we calculated intakes of major food sources of protein (red meat, poultry, seafood including fish and shellfish and soy) based on their relative composition in recipes and mixed dishes.

### 2.3. Assessment of Physical Activity and Anthropometric Measures

Data on participants’ physical activity in metabolic equivalent of task (MET)-hours per week were measured using the SP2 Physical Activity Questionnaire (SP2PAQ), an assessment adapted from several established physical activity questionnaires [29]. The SP2PAQ assessed the type, frequency and duration of activities undertaken during transportation, occupation, leisure time and household work. The SP2PAQ was validated against accelerometer data, and, while reasonably high correlations were observed for vigorous-intensity activity (*r* = 0.73), the correlations for moderate-intensity activity were modest (*r* = 0.27) [29]. Our study focused on combined moderate-to-vigorous physical activity (MVPA) as an exposure, rather than total physical activity, based on established guidelines that highlight this level of intensity as providing the most health benefits [30].

We assessed both body mass index (BMI) and waist circumference as measures of adiposity. BMI was based on weight and height measurements collected during the health examination. Weight was measured kilograms, in light clothing after participants had removed all objects from their pockets, using a portable stadiometer (SECA, model 782-2321009; Vogel & Halke, Hamburg, Germany). Height was measured in centimeters, without shoes and with the head in the Frankfurt Plane position, using the same stadiometer. Waist circumference at the midpoint between the last rib and iliac crest was measured in centimeters, using a stretch-resistant tape.

### 2.4. Assessment of Serum Amino Acids

Fasting serum concentrations of 16 amino acids, namely alanine, arginine, citrulline, glutamate, glutamine, glycine, histidine, isoleucine, leucine, methionine, ornithine, phenylalanine, proline, serine, tyrosine and valine, were measured as the primary outcome of this study. For extraction, 50 µL of serum was spiked with amino acid standards (Sigma Aldrich, St. Louis, MO, USA). The mixture was extracted from samples using methanol, and the extracts were derivatized with 3M hydrochloric acid in butanol (Sigma Aldrich, St. Louis, MO, USA), dried and reconstituted in methanol for analysis in LC-MS. Amino acids were separated using a C18 column (Phenomenex, 100 × 2.1 mm, 1.6 µm, Luna Omega) on an Agilent 1290 Infinity LC system (Agilent Technologies, Santa Clara, CA, USA) coupled with quadrupole-ion trap mass spectrometer (QTRAP 5500, AB Sciex, Washington, DC, USA). Water (mobile phase A) and acetonitrile (mobile phase B), both containing 0.1% formic acid, were used for chromatography separation. Amino acids were ionized in positive mode using electrospray ionization, and the chromatograms were integrated using MultiQuant 3.0.3 software (AB Sciex, Washington, DC, USA). Absolute quantification was done by comparing the ratios of the amino acids to their respective internal standards, against an external calibration curve which consisted of all reported amino acids. Isomeric amino acids glutamine and glutamate were measured as a single peak, as were isoleucine and leucine.

Quality control (QC) was performed on the amino acid measurements prior to statistical analysis. To reduce potential batch effect, the data were log2-transformed and normalized in two steps: first by adjusting the intensity levels of each metabolite according to the QC samples run in each batch and second by equalizing the average log2-concentration across the batches. Internal variation and missing values were not a significant issue, as all amino acid measurements recorded a CV% of less than 20% and a missing rate of less than 5%.

### 2.5. Assessment of Covariates

Covariates in this study included age, sex, ethnicity, smoking status, alcohol consumption and education level, all of which were assessed in the standardized interviewer-administered questionnaire. These covariates were identified a priori based on published literature on sociodemographic and lifestyle determinants of metabolic health [9]. Participants were of Chinese, Malay or Indian ethnicity, which represent the three main ethnic groups in Singapore. Cigarette smoking and alcohol consumption were assessed in terms of never, former and current users. Education level was based on highest level of educational attainment.

### 2.6. Statistical Analysis

For this cross-sectional analysis, associations between potential determinants, including dietary, physical activity and anthropometric variables, and serum amino acid levels were modeled using multiple linear regression. Amino acid levels were standardized to Z-scores to approximate normality, and these standardized values were used for modeling and reporting. Certain amino acids were also combined into two functional groups: the aromatic amino acids (AAA), including phenylalanine and tyrosine, and the BCAA species.

While the intakes of red meat, poultry, seafood and soy were originally measured in grams per day, these as well as total protein were modeled as percentages of total energy (en%) using reported FFQ values for total daily energy intake. The intakes of red meat, poultry and seafood were summed up to calculate the total meat and seafood variable. For consistency across categories, total protein intake and intakes of the dietary protein sources were converted into percentage of total energy and multiplied by 100. BMI, waist circumference and the dietary intakes were modeled as continuous variables. Physical activity was modeled in quartiles of total MVPA rather than continuous MET-hours per week, a common practice in physical activity studies that also aids interpretation of results [30]. We also measured the trend effect associated with moving from the initial MVPA quartile to subsequent quartiles using the Wilcoxon signed-rank test and reported its significance level.

All linear models were adjusted for age (years), sex (male/female), ethnicity (Chinese/Malay/Indian), cigarette smoking (never/former/current), alcohol consumption (never/former/current), education level (none or lower primary/primary/secondary/polytechnic/university) and total energy intake (kilocalories per day). Additionally, the models for dietary variables were adjusted for BMI (kg/m^2^) and MVPA (MET-hours/week); those for MVPA were adjusted for BMI and total protein intake (en%); and those for the anthropometric variables were adjusted for MVPA and total protein intake. Age, BMI, waist circumference and protein intake as a percentage of total energy were approximately normally distributed, while total energy intake, total MVPA and daily intake in grams per day of the dietary protein sources were skewed. We also tested the interaction with sex for associations between the potential determinants and amino acid levels by creating multiplicative interaction terms for sex and each potential determinant (total protein, total meat and fish, red meat, poultry, seafood, soy, MVPA, BMI and waist circumference). If we observed a significant interaction effect, we reported sex-stratified results for both male and female participants, in addition to results for the overall study population. Reported *p*-values are two-sided and are adjusted for multiple testing using the Benjamini–Hochberg procedure to control the false-discovery rate at an alpha-level of 0.05. Adjusted *p*-values below this threshold of 0.05 were deemed to be statistically significant. Analyses were performed using R version 3.5.1 (R Core Team, Vienna, Austria).

## 3. Results

Table 1 shows the characteristics of the study population overall and according to sex. There were slightly more females (*n* = 1707) than males (*n* = 1392) in the overall population, although there was no significant difference in age or ethnic distribution between females and males. The average age of male participants was 46.0 years (SD: 11.5 years), while the average age of female participants was 45.5 years (SD: 11.1 years). The population was mostly ethnic Chinese (75.1% overall), followed by Malay (15.1%) and Indian (10.8%), which is consistent with the demographics of Singapore at the national level [26]. Seafood was the most consumed dietary protein source we evaluated (mean daily intake 80.2 g), followed by poultry (49.6 g), red meat (48.5 g) and soy (14.1 g). While females had significantly higher protein intake as a percentage of total energy than males, males had higher intakes of protein sources in grams per day as their total energy intake was higher. Males also had significantly higher physical activity, BMI and waist circumference compared with females. In addition, a higher proportion of males were current smokers, alcohol drinkers and university-educated compared with females.

### 3.1. Intake of Protein and Its Food Sources

Table 2 shows the associations between dietary protein intake and its major food sources and serum amino acid levels after adjusting for age, sex, ethnicity, smoking and alcohol drinking status, education level, BMI, total MVPA and total energy intake (see Appendix A for results for red meat and poultry; see Appendix A for results for seafood and soy; and see Appendix A for overall results of a simpler model adjusting for age, sex and ethnicity). In the overall study population, intakes of total protein, total meat and seafood and seafood were inversely associated with citrulline levels. In the overall population, a doubling of total protein intake as a percentage of total energy was associated with a 0.103-SD decrease in citrulline concentration, while a doubling of total meat and seafood was associated with a 0.111-SD decrease. Furthermore, the associations between intake of total protein and total meat and seafood and BCAA levels were not statistically significant, although intake of seafood was associated with higher levels of valine and combined BCAA. Likewise, soy intake was associated with higher proline levels in the overall population. We identified several significant interactions with sex for the associations between dietary intakes and serum amino acid levels. Specifically, significant sex interactions were observed for the associations between total protein and total meat and seafood and levels of isoleucine/leucine, valine and combined BCAA. Total protein and total meat and seafood were associated with higher isoleucine/leucine, valine and combined BCAA levels in females, but not in males. Similarly, intake of seafood was associated with higher isoleucine/leucine, valine and combined BCAA levels in females, but not in males. Intakes of red meat and poultry were not significantly associated with amino acid levels, either overall or stratified by sex.

### 3.2. Physical Activity

We did not identify significant interactions with sex for the association between MVPA and serum amino acid levels, and therefore reported multivariable results (Table 3) for the overall study population (see Appendix A for results of the simpler model). Due to the small effect sizes resulting from single MET-hours, MVPA data were modeled in quartiles, with Quartile 1 (total MVPA < 8.75 MET-hours/week) serving as the reference category, up to Quartile 4 (MVPA ≥ 55.00 MET-hours/week). Significant trends were observed for alanine, arginine, glutamate/glutamine, isoleucine/leucine, methionine, proline, serine and combined BCAA, whereby higher MVPA levels were associated with lower serum levels of each of the amino acids.

### 3.3. Anthropometric Measures

Table 4 shows multivariable-adjusted associations between measures of adiposity and serum amino acid levels (see Appendix A for overall results of the simpler model). Because we identified several significant interactions by sex, we show both overall and sex-stratified associations. Both BMI (kg/m^2^) and waist circumference (cm) were modeled continuously. Significant interactions by sex were observed for the associations between BMI and levels of proline and tyrosine, as well as for waist circumference and levels of glutamate/glutamine, ornithine, proline, valine and combined BCAA. Associations were mostly stronger in females than in males, although the association between BMI and tyrosine was stronger in males. Overall, both BMI and waist circumference were associated with higher levels of alanine, glutamate/glutamine, isoleucine/leucine, ornithine, phenylalanine, proline, tyrosine, valine, combined AAA and combined BCAA. Furthermore, BMI and waist circumference were inversely associated with levels of glycine and serine. Additionally, BMI was inversely associated with levels of arginine, citrulline and histidine, while waist circumference was directly associated with levels of methionine.

## 4. Discussion

We investigated potential lifestyle determinants of levels of 16 circulating amino acids in a multiethnic Asian population, including dietary protein intakes, physical activity and measures of adiposity. We identified significant associations between these modifiable factors and several amino acids after multivariable adjustment for a range of potential confounders and correcting for multiple testing. Higher intakes of seafood, total meat and seafood (in females) and total protein (in females); greater adiposity assessed as both BMI and waist circumference; and less MVPA were all associated with higher serum BCAA levels. In addition, these modifiable factors were associated with several other circulating amino acids. Intakes of total protein, total meat and seafood and seafood specifically were inversely associated with citrulline levels and soy intake was directly associated with higher proline levels. Furthermore, higher physical activity levels were inversely associated with alanine, arginine, glutamate/glutamine, methionine, proline and serine levels. Finally, higher adiposity was associated with higher levels of alanine, glutamate/glutamine, ornithine, proline and AAA, as well as inversely associated with arginine, glycine and serine levels.

These results may have important health implications in what they reveal about potential lifestyle determinants of AAA and BCAA levels, as AAA and BCAA species have been identified as risk factors for type 2 diabetes and cardiovascular diseases [13,19,20,21,22,23,31,32,33,34,35,36,37]. Of these species, isoleucine, leucine, phenylalanine, tryptophan and valine are essential amino acids, while tyrosine is conditionally essential. Dietary intake therefore constitutes a major source of these species in humans, and it is not surprising that elevated levels have been linked to higher dietary protein intake. Studies conducted in Brazil, Germany, Poland and Canada have identified strong associations between red meat intake and BCAA levels [14,15,16,17]. In contrast, intake of seafood, but not red meat, was associated with BCAA levels in our study. According to data from the U.S. Department of Agriculture, red meat and poultry generally have a higher content of BCAA per gram as compared with seafood [18]. However, it is worth noting that seafood was the biggest source of dietary protein in our population, with a mean daily consumption of 80 g compared to 49 g for red meat and 50 g for poultry. This is in contrast to the other study populations, in which red meat was reported to be the biggest source of dietary protein [14,15,16,17]. Furthermore, the correlation between red meat and seafood intake in our population was modest (*r* = 0.20 overall, 0.18 in males, 0.22 in females), suggesting our results were not confounded by correlations between different dietary protein sources. It is possible that BCAA levels reflect intake of total dietary protein or animal protein and that the dominant food sources in a specific population will correlate most strongly with BCAA levels. This is supported by a previous meta-analysis which reported an attenuation of the association between BCAA concentrations and diabetes risk after adjusting for total protein intake, indicating total protein intake, regardless of dietary source, is a key determinant of circulating BCAA levels [18]. However, further studies in populations with low red meat consumption relative to other dietary protein sources are warranted to confirm these results.

A notable finding of our study was the significant interactions with sex for associations between certain lifestyle determinants and amino acid levels, particularly with regards to dietary protein intake and BCAA levels. We observed significant interactions with sex for the associations between total protein and total meat and fish intake and BCAA levels. These interactions may have been responsible for the markedly different associations between these dietary intakes and BCAA levels in females compared with males, whereby the associations were significantly stronger in females than in males. Furthermore, while we did not detect significant interactions with sex for the associations between total protein intake and levels of other amino acids, these associations were generally stronger in females compared with males (Table 2). Additionally, we observed significant interactions with sex for associations between adiposity and levels of glutamate/glutamine, ornithine, proline, tyrosine and BCAA. Each of these associations was stronger in females than in males, except for the association between BMI and tyrosine, which was stronger in males. To our knowledge, this is the first study to report significant interactions with sex for the associations between lifestyle factors and amino acid levels, and, while sex differences in the associations for adiposity may be biologically related to the differential distribution of body fat in females and males, an interaction between sex and dietary protein intake is more difficult to explain biologically and requires confirmation.

Few studies have investigated the role of physical activity as a determinant of serum amino acid levels. It is generally understood that exercise promotes BCAA oxidation through activation of the branched-chain *α*-keto acid dehydrogenase (BCKDH) complex, which catalyzes the BCAA catabolic pathway [38]. This mechanism is consistent with our observation of an inverse association between physical activity and BCAA levels. In contrast, in a clinical trial comparing plasma BCAA concentrations in overweight, insulin-resistant participants before and after a six-month regimen of vigorous-intensity aerobic and resistance training, no reduction in BCAA levels was observed following the intervention [39]. It is difficult to directly compare the results of this study and our own, given the differences in study design and participant populations. The metabolic status of the trial participants may have modified their ability to clear circulating BCAA via the BCKDH pathway, and our results may therefore be more representative of a metabolically healthy cohort. That being said, an analysis of serum metabolite networks in the EPIC-Potsdam cohort, a more comparable group to ours in terms of metabolic health status, reported levels of a cluster of amino acids, including the BCAA species, to be directly associated with objectively-measured physical activity energy expenditure [25]. This again seems to conflict with our findings and the biological understanding of the BCKDH pathway. Further studies into the association between MVPA and BCAA levels are warranted to clarify these inconsistencies.

Likewise, adiposity has not often been the main focus of studies of determinants of amino acid levels, with investigators typically considering it as a covariate in multivariable models rather than as the main determinant. A previous study in the Singapore population reported associations between a higher waist-to-hip ratio and higher levels of alanine, histidine, methionine, phenylalanine and tyrosine, which is partially consistent with our results [40]. However, they did not observe significant associations with the BCAA species, nor did they investigate potential interactions by sex. Our results are also partially consistent with those of an analysis in the EPIC-Oxford cohort, which found BMI to be directly associated with serum levels of isoleucine and tyrosine, but not associated with phenylalanine or valine [23]. Consistent with our findings, levels of all three BCAA species were higher [41], and glycine levels lower [42], in obese participants compared with lean participants in U.S., European and Japanese studies. Furthermore, increased MVPA has long been recommended to reduce adverse health effects of obesity [43], and biomarkers reflective of physical activity level may display opposite associations with adiposity. This is generally what we observed in our study, as AAA and BCAA levels were inversely associated with MVPA and directly associated with adiposity. The exceptions were arginine and serine, which were inversely associated with both MVPA and adiposity.

In addition to AAA and BCAA species, several other amino acids were significantly associated with lifestyle and anthropometric determinants in our study. Although some of these amino acids have also been previously linked to adverse health outcomes, they have generally not been examined in relation to lifestyle determinants. Levels of alanine and combined glutamate/glutamine were inversely associated with MVPA and directly associated with adiposity in our study. These species are produced from amino acid catabolism in the muscle where BCAA species serve as nitrogen donors, and it is possible that their serum levels reflect exposures to the same lifestyle determinants of BCAA levels [32]. Additionally, the direct association with adiposity is consistent with these species’ role in gluconeogenesis, as higher adiposity is associated with elevated fasting glucose and increased gluconeogenesis. Alanine and glutamate both stimulate glucagon secretion [44,45], and alanine in particular is dysregulated in nonalcoholic fatty liver disease [44]. In addition, levels of ornithine and proline were directly associated with adiposity, while proline was also directly associated with soy intake and inversely associated with MVPA in our study. Ornithine and proline share a common metabolic pathway, and increased proline levels can indicate decreased bioavailability of nitric oxide and may contribute to metabolic complications such as insulin resistance and diabetes [46]. Levels of glycine were inversely associated with adiposity. Glycine has been suggested to provide protection against diabetes based on identification of genetic instruments, reflecting higher levels of glycine and associated with lower diabetes risk in Mendelian randomization studies [47]. Levels of methionine were inversely associated with MVPA and directly associated with waist circumference in our study. Methionine is involved in homocysteine metabolism, and the dysregulation of this pathway may contribute to atherosclerosis [48]. Finally, levels of citrulline were inversely associated with BMI and intakes of total protein, total meat and seafood and seafood specifically. This is somewhat consistent with a previous study that reported higher citrulline concentrations in vegetarians and vegans as compared with meat eaters [11]. Citrulline serves as an arginine precursor and has been identified as a biomarker of intestinal injury, with lower concentrations associated with greater intestinal disease severity [49]. These results have shed some light on significant lifestyle determinants of levels of clinically important amino acids, but further studies are warranted to clarify the potential health impacts of these species.

Taken together, our results suggest a lifestyle with lower meat and seafood consumption, more moderate-to-vigorous intensity physical activity and avoidance of excess adiposity is associated with lower levels of BCAA species. Lower BCAA concentrations achieved through lifestyle modifications over a prolonged period may in turn result in lower risks of type 2 diabetes [20,22,31,32], coronary artery disease and heart failure [21,33,34]. Evidence from Mendelian randomization studies support a causal role for BCAA species in diabetes development [23]. Therefore, our results suggest lifestyle patterns marked by lower dietary protein intake, higher amounts of physical activity and lower adiposity may be associated with lower cardiometabolic disease risk. These lifestyle patterns of following a healthy diet, getting sufficient physical exercise and managing body weight are generally consistent with published guidelines for prevention of cardiometabolic diseases [3,4], although our study also highlights the potential contribution of high seafood consumption to elevated BCAA levels. Seafood is typically promoted as a healthier alternative to red meat, but our findings raise the possibility that higher consumption of seafood may also have detrimental effects in populations where seafood intake is already high. However, further studies, preferably including randomized trials, are required to confirm this finding.

A key strength of our study is the large number of ethnic Asian participants, which allowed us to analyze the impact of dietary intake patterns in a population where red meat is not the major dietary protein source. Our study also has several potential limitations. While we were able to perform stratified analyses and test for interactions by sex, our study population did not include sufficient numbers of ethnic Malay and Indian participants for us to detect potential ethnic differences. Serum amino acid data were available only for SP2 participants who had attended the health examination. An analysis of factors associated with higher participation in the health examination reported that older age, Chinese and Indian ethnicity, higher education level and greater physical activity were key predisposing factors [50], which raises the possibility that our results may not be as valid in populations that do not fit this profile. Data on dietary intake and MVPA were collected using questionnaires, which are commonly used in epidemiological studies and were reasonably accurate based on our validation studies but remain imperfect measurement tools. Measurement error in the assessment of our exposure variables is most likely to have led to weaker associations between lifestyle factors and amino acid levels. Our use of the anthropometric measures BMI and waist circumference, as opposed to more direct measures of body fatness such as body composition, may have also resulted in measurement error and an underestimation of the association between adiposity and amino acid levels. Similarly, our targeted metabolomics protocol for measuring amino acid concentrations was not exhaustive, and several species with potentially important clinical implications, such as lysine and individually-measured glutamate and glutamine, were not included. Furthermore, our cross-sectional study does not allow us to establish the direction of cause and effect, although participants will not have been aware of their amino acid levels, making an effect of these metabolite levels on lifestyle less likely. Additionally, although we adjusted for a range of confounders in our multivariable models, residual confounding remained a potential source of bias in this study. Finally, our results apply to a multiethnic Asian population and may not necessarily generalize to other populations.

In conclusion, our study identified several dietary factors, physical activity and adiposity as potentially important determinants of levels of BCAA species and a range of other circulating amino acids. Notably, our findings highlight the heretofore understudied association between physical activity and measures of adiposity with these metabolites. Future studies in multiethnic populations could highlight potential ethnic differences, while further studies focusing on modifiable determinants of amino acid levels are warranted given the increasing evidence for the role of several amino acids in the development of cardiometabolic disease.

## Figures and Tables

**Figure 1 nutrients-12-02603-f001:**
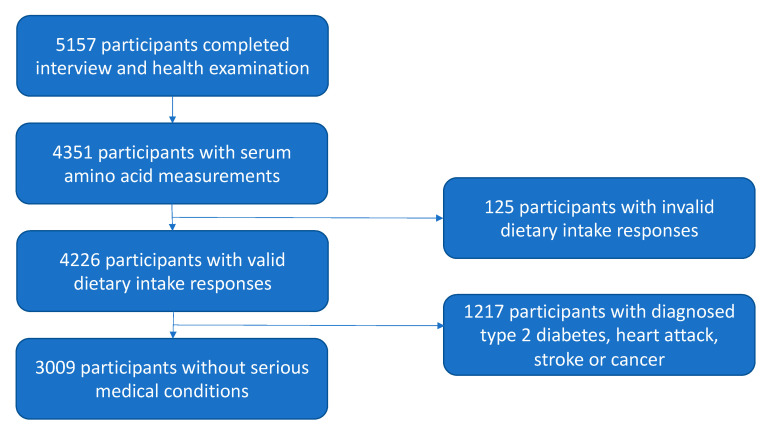
Flow chart describing the selection process for the study population.

**Table 1 nutrients-12-02603-t001:** Characteristics of the study population overall and by sex.

	Overall (*n* = 3099)	Males (*n* = 1392)	Females (*n* = 1707)	*p*-Value ^1^
Age (years) Mean (SD)	45.7 (11.3)	46.0 (11.5)	45.5 (11.1)	0.214 *
Ethnicity				
No. Chinese (%)	2298 (75.1)	1037 (74.5)	1261 (73.9)	0.237 ^#^
No. Malays (%)	467 (15.1)	205 (14.7)	262 (15.3)	
No. Indians (%)	334 (10.8)	150 (10.8)	184 (10.8)	
Total energy intake (kcal/day) Median (IQR)	1992 (956)	2328 (1046)	1828 (910)	<0.001 ^^^
Dietary intake Total protein intake (% total energy/day) Mean (SD)	15.5 (2.3)	15.2 (2.2)	15.7 (2.3)	<0.001 *
Total red meat intake (g/day) Mean (SD)	48.5 (47.6)	54.4 (51.6)	43.8 (43.5)	<0.001 ^^^
Total poultry intake (g/day) Mean (SD)	49.6 (40.2)	55.8 (42.0)	44.6 (37.9)	<0.001 ^^^
Total seafood intake (g/day) Mean (SD)	80.2 (60.2)	83.9 (58.3)	77.1 (61.6)	0.002 ^^^
Total soy intake (g/day) Mean (SD)	14.1 (14.3)	14.9 (14.4)	13.4 (14.1)	0.005 ^^^
Total moderate-to-vigorous physical activity (MET-hours/week) Median (IQR)	22.8 (46.2)	28.0 (67.2)	19.3 (33.8)	<0.001 ^^^
BMI (kg/m^2^) Mean (SD)	23.4 (4.1)	23.8 (3.8)	23.1 (4.3)	<0.001 *
Waist circumference (cm) Mean (SD)	82.2 (11.6)	87.5 (10.9)	77.9 (10.4)	<0.001 *
Smoking status				<0.001 ^#^
No. Never (%)	2483 (80.1)	860 (61.8)	1623 (95.1)	
No. Former (%)	245 (7.9)	205 (14.7)	40 (2.3)	
No. Current (%)	371 (12.0)	327 (23.5)	44 (2.6)	
Drinking status				<0.001 ^#^
No. Never (%)	1817 (58.6)	645 (46.3)	1172 (68.7)	
No. Former (%)	477 (15.4)	227 (16.3)	250 (14.6)	
No. Current (%)	805 (26.0)	520 (37.4)	285 (16.7)	
Education level				<0.001 ^#^
No. None/Lower primary (%)	150 (4.8)	41 (3.0)	109 (6.4)	
No. Primary (%)	518 (16.7)	194 (13.9)	324 (19.0)	
No. Secondary (%)	1272 (41.1)	547 (39.3)	725 (42.5)	
No. Polytechnic (%)	572 (18.5)	273 (19.6)	299 (17.5)	
No. University (%)	587 (18.9)	337 (24.2)	250 (14.6)	

^1^*p*-values compare males and females; * *p*-value calculated using Student’s *t*-test; ^#^*p*-value calculated using the *χ*^2^ test; ^^^
*p*-value calculated using the Kruskal–Wallis test.

**Table 2 nutrients-12-02603-t002:** Associations ^1^ between dietary protein intake ^2^ and serum amino acid levels ^3^. Significant associations following adjustment for multiple testing (adjusted *p* < 0.05) are in bold.

	Total Protein (Energy %)			Total Meat and Seafood (Energy %)		
	Overall	Males	Females	Overall	Males	Females
Alanine *β*-coefficient (95% CI)	−0.045 (−0.143, 0.054)	−0.110 (−0.265, 0.045)	0.009 (−0.120, 0.138)	−0.050 (−0.149, 0.048)	−0.116 (−0.269, 0.038)	0.003 (−0.126, 0.131)
Arginine *β*-coefficient (95% CI)	0.024 (−0.078, 0.125)	0.018 (−0.143, 0.179)	0.032 (−0.101, 0.164)	0.017 (−0.084, 0.119)	0.018 (−0.142, 0.178)	0.019 (−0.114, 0.151)
Citrulline *β*-coefficient (95% CI)	**−0.103** **(−0.198, −0.007)**	−0.124 (−0.275, 0.028)	−0.084 (−0.207, 0.038)	**−0.111** **(−0.205, −0.016)**	−0.131 (−0.282, 0.019)	−0.094 (−0.217, 0.029)
Glutamate/Glutamine *β*-coefficient (95% CI)	−0.014 (−0.108, 0.081)	−0.089 (−0.243, 0.066)	0.055 (−0.063, 0.174)	−0.005 (−0.099, 0.089)	−0.078 (−0.231, 0.076)	0.060 (−0.059, 0.178)
Glycine *β*-coefficient (95% CI)	−0.065 (−0.166, 0.035)	−0.107 (−0.237, 0.023)	−0.009 (−0.156, 0.139)	−0.079 (−0.179, 0.021)	−0.107 (−0.236, 0.022)	−0.033 (−0.181, 0.115)
Histidine *β*-coefficient (95% CI)	−0.044 (−0.146, 0.059)	−0.015 (−0.175, 0.146)	−0.055 (−0.187, 0.077)	−0.051 (−0.153, 0.051)	−0.017 (−0.177, 0.143)	−0.070 (−0.202, 0.062)
Isoleucine/Leucine * *β*-coefficient (95% CI)	0.048 (−0.038, 0.134)	−0.040 (−0.183, 0.103)	**0.123** **(0.018, 0.229)**	0.055 (−0.031, 0.141)	−0.025 (−0.168, 0.117)	**0.123** **(0.018, 0.228**)
Methionine *β*-coefficient (95% CI)	0.041 (−0.056, 0.138)	0.060 (−0.010, 0.219)	0.039 (−0.082, 0.160)	0.050 (−0.047, 0.147)	0.072 (−0.086, 0.231)	0.044 (−0.077, 0.164)
Ornithine *β*-coefficient (95% CI)	0.036 (−0.060, 0.132)	−0.018 (−0.171, 0.135)	0.097 (−0.027, 0.220)	0.023 (−0.073, 0.119)	−0.029 (−0.181, 0.123)	0.082 (−0.042, 0.205)
Phenylalanine *β*-coefficient (95% CI)	0.045 (−0.053, 0.142)	0.060 (−0.099, 0.218)	0.034 (−0.088, 0.157)	0.048 (−0.049, 0.145)	0.069 (−0.089, 0.226)	0.031 (−0.092, 0.154)
Proline *β*-coefficient (95% CI)	−0.042 (−0.139, 0.056)	−0.125 (−0.278, 0.027)	0.023 (−0.104, 0.150)	−0.061 (−0.158, 0.036)	−0.144 (−0.295, 0.008)	0.003 (−0.123, 0.130)
Serine *β*-coefficient (95% CI)	0.052 (−0.049, 0.153)	0.035 (−0.113, 0.183)	0.065 (−0.075, 0.206)	0.041 (−0.060, 0.142)	0.038 (−0.109, 0.185)	0.044 (−0.096, 0.185)
Tyrosine *β*-coefficient (95% CI)	0.033 (−0.065, 0.131)	−0.025 (−0.181, 0.131)	0.074 (−0.051, 0.199)	0.042 (−0.056, 0.139)	−0.013 (−0.168, 0.142)	0.078 (−0.048, 0.203)
Valine * *β*-coefficient (95% CI)	0.057 (−0.035, 0.148)	−0.066 (−0.211, 0.080)	**0.159** **(0.043, 0.276)**	0.067 (−0.024, 0.158)	−0.047 (−0.191, 0.098)	**0.162** **(0.045, 0.278)**
Aromatic *β*-coefficient (95% CI)	0.011 (−0.088, 0.110)	0.004 (−0.153, 0.161)	0.020 (−0.108, 0.147)	0.013 (−0.086, 0.111)	0.011 (−0.145, 0.168)	0.014 (−0.114, 0.141)
Branched-chain * *β*-coefficient (95% CI)	0.055 (−0.033, 0.144)	−0.058 (−0.201, 0.086)	**0.150** **(0.039, 0.261)**	0.064 (−0.024, 0.152)	−0.040 (−0.182, 0.103)	**0.151** **(0.040, 0.263)**

^1^ Adjusted for age, sex, ethnicity, body mass index, smoking status, alcohol drinking status, moderate-to-vigorous physical activity and education level. ^2^
*β*-coefficients correspond to the change in amino acid levels associated with a 100% increase, or doubling, in daily intake of a given dietary source. ^3^ Amino acid concentrations were converted into *Z*-scores and are expressed per-SD. * Significant interaction by sex on the association between dietary protein intakes and these amino acids.

**Table 3 nutrients-12-02603-t003:** Associations ^1^ between quartiles of moderate-to-vigorous physical activity ^2^ and serum amino acid levels ^3^ using the lowest quartile as the reference category. Significant trends following adjustment for multiple testing (adjusted *p* < 0.05) are in bold.

	Quartile 1 ^4^ (*n* = 776)	Quartile 2 ^5^ (*n* = 774)	Quartile 3 ^6^ (*n* = 774)	Quartile 4 ^7^ (*n* = 775)	*p*-Value for Trend ^8^
Alanine *β*-coefficient (95% CI)	0.000 (reference)	−0.067 (−0.163, 0.028)	−0.090 (−0.186,0.006)	−0.203 (−0.301, −0.106)	**<0.001**
Arginine *β*-coefficient (95% CI)	0.000 (reference)	0.002 (−0.097, 0.100)	−0.150 (−0.249, −0.051)	−0.063 (−0.164, 0.037)	**0.033**
Citrulline *β*-coefficient (95% CI)	0.000 (reference)	−0.002 (−0.095, 0.090)	0.020 (−0.073, 0.113)	0.030 (−0.064, 0.124)	0.467
Glutamate/Glutamine *β*-coefficient (95% CI)	0.000 (reference)	−0.055 (−0.146, 0.037)	−0.120 (−0.212,−0.028)	−0.119 (−0.212, −0.026)	**0.005**
Glycine *β*-coefficient (95% CI)	0.000 (reference)	0.067 (−0.030, 0.165)	0.006 (−0.092, 0.104)	0.022 (−0.077, 0.121)	0.961
Histidine *β*-coefficient (95% CI)	0.000 (reference)	0.060 (−0.039,0.160)	0.046 (−0.053, 0.146)	−0.042 (−0.143, 0.058)	0.414
Isoleucine/Leucine *β*-coefficient (95% CI)	0.000 (reference)	−0.019 (−0.103, 0.065)	−0.036 (−0.120, 0.048)	−0.116 (−0.202, −0.031)	**0.008**
Methionine *β*-coefficient (95% CI)	0.000 (reference)	−0.008 (−0.102, 0.086)	−0.060 (−0.155, 0.034)	−0.110 (−0.205, −0.014)	**0.014**
Ornithine *β*-coefficient (95% CI)	0.000 (reference)	−0.063 (−0.156, 0.031)	−0.077 (−0.171, 0.017)	0.002 (−0.093, 0.097)	0.916
Phenylalanine *β*-coefficient (95% CI)	0.000 (reference)	−0.011 (−0.105, 0.084)	−0.027 (−0.122, 0.068)	−0.087 (−0.183, 0.009)	0.077
Proline *β*-coefficient (95% CI)	0.000 (reference)	−0.066 (−0.160, 0.029)	−0.132 (−0.226, −0.037)	−0.186 (−0.282, −0.091)	**<0.001**
Serine *β*-coefficient (95% CI)	0.000 (reference)	−0.024 (−0.123, 0.074)	−0.091 (−0.190, 0.008)	−0.136 (−0.236, −0.036)	**0.003**
Tyrosine *β*-coefficient (95% CI)	0.000 (reference)	−0.029 (−0.124, 0.066)	−0.088 (−0.183, 0.007)	−0.074, (−0.170, 0.023)	0.070
Valine *β*-coefficient (95% CI)	0.000 (reference)	0.008 (−0.081, 0.096)	−0.026 (−0.114, 0.063)	−0.076 (−0.166, 0.014)	0.076
Aromatic *β*-coefficient (95% CI)	0.000 (reference)	0.010 (−0.086, 0.107)	−0.027 (−0.124, 0.069)	−0.081 (−0.179, 0.017)	0.079
Branched-chain *β*-coefficient (95% CI)	0.000 (reference)	−0.003 (−0.089, 0.083)	−0.031 (−0.117, 0.055)	−0.096 (−0.183, −0.008)	**0.002**

^1^ Adjusted for age, sex, ethnicity, body mass index, smoking status, alcohol drinking status, total protein intake (en%) and education level. ^2^
*β*-coefficients correspond to the change in amino acid levels associated with moving from the first MVPA quartile to successive quartiles. ^3^ Amino acid concentrations were converted to *Z*-scores and are expressed per-SD. ^4^ Quartile 1: <8.75 MET-hours/week. ^5^ Quartile 2: 8.75–22.74 MET-hours/week. ^6^ Quartile 3: 22.75–54.99 MET-hours/week. ^7^ Quartile 4: ≥55.00 MET-hours/week. ^8^
*p*-values calculated using the Wilcoxon signed-rank test.

**Table 4 nutrients-12-02603-t004:** Associations ^1^ between anthropometric measures ^2^ and serum amino acid levels ^3^. Significant associations following adjustment for multiple testing (adjusted *p* < 0.05) are in bold.

	Body Mass Index (kg/m^2^)			Waist Circumference (cm)		
	Overall	Males	Females	Overall	Males	Females
Alanine *β*-coefficient (95% CI)	**0.036** **(0.027, 0.045)**	**0.031** **(0.017, 0.045)**	**0.039** **(0.027, 0.050)**	**0.020** **(0.017, 0.023)**	**0.013** **(0.008, 0.018)**	**0.017** **(0.013, 0.022)**
Arginine *β*-coefficient (95% CI)	**−0.014** **(−0.023, −0.005)**	**−0.011** **(−0.025, −0.003)**	**−0.016** **(−0.028, −0.004)**	0.000 (−0.003, 0.004)	0.000 (−0.005, 0.005)	−0.007 (−0.011, −0.002)
Citrulline *β*-coefficient (95% CI)	**−0.019** **(−0.027, −0.010)**	−0.012 (−0.025, 0.002)	**−0.023** **(−0.034, −0.012)**	0.001 (−0.002, 0.004)	−0.005 (−0.010, 0.000)	**−0.007** **(−0.012, −0.003)**
Glutamate/Glutamine *β*-coefficient (95% CI)	**0.063** **(0.054, 0.071)**	**0.060** **(0.046, 0.074)**	**0.063** **(0.052, 0.073)**	**0.031** **(0.028, 0.034)**	**0.021** **(0.016, 0.026)**	**0.027** **(0.023, 0.032)**
Glycine *β*-coefficient (95% CI)	**−0.046** **(−0.055, −0.037)**	**−0.050** **(−0.061, −0.038)**	**−0.049** **(−0.063, −0.036)**	**−0.017** **(−0.020, −0.013)**	**−0.016** **(−0.020, −0.012)**	**−0.018** **(−0.024, −0.013)**
Histidine *β*-coefficient (95% CI)	**−0.013** **(−0.022, −0.004)**	−0.016 (−0.031, −0.002)	−0.014 (−0.026, −0.002)	0.002 (−0.001, 0.006)	−0.004 (−0.009, 0.001)	−0.001 (−0.006, 0.004)
Isoleucine/Leucine *β*-coefficient (95% CI)	**0.056** **(0.049, 0.064)**	**0.060** **(0.047, 0.073)**	**0.050** **(0.041, 0.060)**	**0.034** **(0.031, 0.037)**	**0.020** **(0.016, 0.026)**	**0.021** **(0.017, 0.025)**
Methionine *β*-coefficient (95% CI)	0.008 (−0.001, 0.017)	0.011 (−0.003, 0.026)	0.003 (−0.007, 0.014)	**0.014** **(0.010, 0.017)**	0.006 (0.001, 0.011)	0.001 (−0.003, 0.006)
Ornithine *β*-coefficient (95% CI)	**0.018** **(0.009, 0.027)**	0.014 (0.001, 0.028)	**0.018** **(0.007, 0.029)**	**0.015** **(0.012, 0.018)**	0.003 (−0.002, 0.008)	**0.010** **(0.005, 0.014)**
Phenylalanine *β*-coefficient (95% CI)	**0.045** **(0.037, 0.054)**	**0.047** **(0.033, 0.061)**	**0.043** **(0.032, 0.054)**	**0.022** **(0.019, 0.025)**	**0.013** **(0.008, 0.018)**	**0.018** **(0.013, 0.022)**
Proline * *β*-coefficient (95% CI)	**0.026** **(0.017, 0.034)**	0.006 (−0.007, 0.020)	**0.039** **(0.027, 0.050)**	**0.019** **(0.015, 0.022)**	0.004 (−0.001, 0.009)	**0.018** **(0.013, 0.022)**
Serine *β*-coefficient (95% CI)	**−0.029** **(−0.038, −0.020)**	**−0.031** **(−0.044, −0.018)**	**−0.027** **(−0.040, −0.015)**	**−0.011** **(−0.015, −0.008)**	**−0.010** **(−0.015, −0.006)**	**−0.010** **(−0.016, −0.005)**
Tyrosine *β*-coefficient (95% CI)	**0.061** **(0.052, 0.069)**	**0.071** **(0.057, 0.084)**	**0.053** **(0.041, 0.064)**	**0.025** **(0.022, 0.028)**	**0.023** **(0.019, 0.028)**	**0.021** **(0.016, 0.026)**
Valine *β*-coefficient (95% CI)	**0.062** **(0.054, 0.070)**	**0.059** **(0.046, 0.073)**	**0.061** **(0.050, 0.071)**	**0.032** **(0.029, 0.035)**	**0.018** **(0.014, 0.023)**	**0.027** **(0.022, 0.031)**
Aromatic *β*-coefficient (95% CI)	**0.037** **(0.028, 0.046)**	**0.040** **(0.026, 0.054)**	**0.032** **(0.021, 0.044)**	**0.020** **(0.016, 0.023)**	**0.013** **(0.008, 0.018**)	**0.015** **(0.010, 0.020)**
Branched-chain *β*-coefficient (95% CI)	**0.062** **(0.054, 0.070)**	**0.062** **(0.049, 0.075)**	**0.058** **(0.048, 0.069)**	**0.034** **(0.031, 0.037)**	**0.020** **(0.015, 0.024)**	**0.025** **(0.021, 0.029)**

^1^ Adjusted for age, sex, ethnicity, smoking status, alcohol drinking status, total protein intake (en%), moderate-to-vigorous physical activity and education level. ^2^
*β*-coefficients correspond to the change in amino acid levels associated with a one-unit increment in body mass index (kg/m^2^) and waist circumference (cm). ^3^ Amino acid concentrations were converted to *Z*-scores and are expressed per-SD. * Significant interaction by sex on the association between adiposity and these amino acids.

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
