# Peer review of "Diet, Physical Activity and Adiposity as Determinants of Circulating Amino Acid Levels in a Multiethnic Asian Population"

_nutrients, 2020, doi:10.3390/nu12092603_

Round 1
Reviewer 1 Report
The study conducted by Gunther et al. reports the results related to food proteins, physical activity and adiposity with the serum concentration of 16 amino acids out of 3099 participants in SP2. The introduction has been really clear, some significant improvements and clarifications should be made in the methods and sections of the results and some tables and the coefficient are not clear for readers.
A few important issues
Materials and methods
1) Why were there no fasting serum amino acid concentrations for 6000 participants? Is it possible to describe differences and similarities with the subset with the serum data? Or is this a selected subset completely different from the other?
2) It may be easier for a reader to have a flow chart of the sample selection process.
3) On line 77/78 it is possible to add some references on "Data on the composition of the food of the Ministry of Health, Singapore".
4) Have the results on the validity of the questionnaires (pages 2 line 81-85 and line 94-96) already been published? If yes, edit the text and add references. If not, move them to the results section.
5) I think that in materials and methods it would be easier to have a subparagraph with "Statistical analysis"
6) Regarding statistical analysis, how did you choose the covariates? Did you apply some methods for covariates with skewed distribution? And when you found a significant interaction, did you stratify? If yes, clarify it in the methods section.
7) In line 161-162 it is not clear what you mean by "... to set the false discovery rate to an alpha level of 0.05". Please specify
8) Did you also plan univariate analyzes before linear regression?
results
9) From table 1 it seems not only the comparison between males and females is different but it could also be an effect of etinicity. Have you tried to stratify the analysis also by ethnicity?How could you explain this "effect"?
10) Table 2 is very difficult to read. Even the comments in the results section were not easy for readers. I think there are many coefficients and it is easy to lose attention and meaning /values. It is better also to write an example to clarify the meaning of the coefficients.
11) In the sentence on page 7 line 216 describe the trend in the quartile of physical activity. It should also be described in more detail in the method section.
Discussion
12) As for the discussion it would be easier to have clearer tables from the results section in order to analyze and revised it.
Minor revision
I) Remove "-" after (MET) page 2 line 90
II) Please, since the Q values ​​are reported for the first time on page 4, line 160 shows the meaning.
Author Response
Reviewer 1
The study conducted by Gunther et al. reports the results related to food proteins, physical activity and adiposity with the serum concentration of 16 amino acids out of 3099 participants in SP2. The introduction has been really clear, some significant improvements and clarifications should be made in the methods and sections of the results and some tables and the coefficient are not clear for readers.
A few important issues
Materials and methods
1) Why were there no fasting serum amino acid concentrations for 6000 participants? Is it possible to describe differences and similarities with the subset with the serum data? Or is this a selected subset completely different from the other?
Response: We thank the reviewer for raising this important point. It would have been clearer for readers had we clarified that serum samples were only available for participants who attended both the health examination and the SP2 interview. Of the 10,445 SP2 eligible SP2 participants, only 5,157 attended the health examination. The subset of 4,351 participants from which amino acid concentrations were measured was a representative subset of the 5,157 health examination attendees. However, a subsequent analysis found that older age, Chinese and Indian ethnicity, higher education level, greater intake of monounsaturated fat, and greater physical activity were factors associated with higher participation in the health examination, so it is possible that attendees may have differed in these ways from non-attendees. This aspect of participant selection has been clarified in the Study Population and Design subsection of the Methods section (page 2, lines 71-72), while a discussion of the implications of the potential differences between attendees and non-attendees has been added to the limitations section of the Discussion (page 13, lines 484-488).
2) It may be easier for a reader to have a flow chart of the sample selection process.
Response: We thank the reviewer for this suggestion and have included a new figure, Figure 1, detailing the selection process for our study population (page 3, line 104).
3) On line 77/78 it is possible to add some references on "Data on the composition of the food of the Ministry of Health, Singapore".
Response: We have added an additional reference for the Singapore Health Promotion Board’s Energy and Nutrient Composition of Food database (page 3, line 112).
4) Have the results on the validity of the questionnaires (pages 2 line 81-85 and line 94-96) already been published? If yes, edit the text and add references. If not, move them to the results section.
Response: We thank the reviewer for raising this point. The results on the validity of both the food frequency questionnaire and the physical activity questionnaire have been published. We have clarified this in the Methods section by adding additional references for these published results (page 3, lines 113, 119, and 131).
5) I think that in materials and methods it would be easier to have a subparagraph with "Statistical analysis"
Response: We thank the reviewer for raising this point and agree the Methods section would be improved with subparagraphs. We have broken the Methods section down into Study Population and Design (page 2, line 64), Assessment of Dietary Intakes (page 3, line 105), Assessment of Physical Activity and Anthropometric Measures (page 3, line 124), Assessment of Serum Amino Acids (page 4, line 145), Assessment of Covariates (page 4, line 167), and Statistical Analysis (page 4, line 175) subsections for greater clarity.
6) Regarding statistical analysis, how did you choose the covariates? Did you apply some methods for covariates with skewed distribution? And when you found a significant interaction, did you stratify? If yes, clarify it in the methods section.
Response: Covariates were selected a priori based on previous literature of sociodemographic and lifestyle determinants of metabolic health, a point that has been added to the Assessment of Covariates subsection of the Methods section (page 4, lines 170-171). Most of our numerical covariates (age, BMI, waist circumference, and protein intake as a percentage of total energy) were approximately normally distributed, while energy intake and moderate-to-vigorous intensity physical activity were skewed. Linear regression models do not assume a normal distribution for covariates or independent variables, so we did not apply any further transformation to the skewed covariates. The amino acid concentrations were also approximately normally distributed after being standardized to Z-scores, a point that is mentioned in the Statistical Analysis subsection of the Methods section (page 4, lines 178-179). Furthermore, we identified significant interactions with sex for the associations between dietary protein intake and the adiposity, and the serum amino acid concentrations. In these cases, we stratified our results by sex and reported results for both male and female participants. This point has been clarified in the Statistical Analysis subsection of the Methods section (page 5, lines 204-206).
7) In line 161-162 it is not clear what you mean by "... to set the false discovery rate to an alpha level of 0.05". Please specify
Response: The Benjamini-Hochberg procedure is a method to address the issue of multiple testing, in which assessing multiple comparisons simultaneously (in our case, analyzing multiple amino acids) increases the likelihood of a chance finding being considered statistically significant. Essentially, the Benjamini-Hochberg procedure establishes a stricter cutoff for significance levels to be statistically significant. We have clarified this in the Statistical Analysis subsection of the Methods section (page 5, lines 207-208).
8) Did you also plan univariate analyzes before linear regression?
Response: We did conduct univariate analyses, but given the high likelihood of residual confounding in this study, we did not report results from univariate models. All of the significant associations we observed using our multivariable models were also significant in univariate models, while glycine (beta: -0.243, 95% CI: -0.440, -0.046) and proline (beta: -0.257, 95% CI: -0.454, -0.060) were also inversely associated with red meat intake.
results
9) From table 1 it seems not only the comparison between males and females is different but it could also be an effect of etinicity. Have you tried to stratify the analysis also by ethnicity?How could you explain this "effect"?
Response: We thank the reviewer for raising this possibility. While the numbers of males and females in our study provided us with sufficient statistical power to conduct stratified analyses by sex, the relatively low numbers of Malay and Indian participants prevented us from doing the same stratification by ethnicity. For example, in ethnic Malay participants, arginine levels were significantly associated with soy intake (beta: 1.766, 95% CI: 0.173, 3.359), but the confidence intervals were extremely wide and unreliable. Results for ethnic Chinese, who made up the majority of our study population, were consistent with the results for the overall population. We agree it would have been interesting to conduct stratified analyses by ethnicity, but our study was not powered to do so.
10) Table 2 is very difficult to read. Even the comments in the results section were not easy for readers. I think there are many coefficients and it is easy to lose attention and meaning /values. It is better also to write an example to clarify the meaning of the coefficients.
Response: We thank the reviewer for suggesting these improvements to Table 2, and have incorporated them into a redesigned table (page 7, lines 249-267). We removed the Q-value for interaction from the table and instead indicate which amino acids saw significant interactions by sex with an asterisk. Furthermore, we entirely removed the beta-values and Q-values from the text of the Results section (page 7, lines 245-246) and also added a description of how to interpret the Table 2 results using citrulline concentrations as an example (page 6, lines 234-237). For the sake of consistency, we made the same changes in design to Table 4 (page 9, lines 309-316).
11) In the sentence on page 7 line 216 describe the trend in the quartile of physical activity. It should also be described in more detail in the method section.
Response: We thank the reviewer for raising this point. In the Statistical Analysis subsection of the Methods section we have clarified that the Q-value for trend reported in our results for physical activity represents the trend effect of moving from the initial MVPA quartile to subsequent quartiles (page 5, lines 191-192).
Discussion
12) As for the discussion it would be easier to have clearer tables from the results section in order to analyze and revised it.
Response: We agree that the Discussion section is easier to read in the context of the new, clearer layout of our tables.
Minor revision
- I) Remove "-" after (MET) page 2 line 90
- II) Please, since the Q values ​​are reported for the first time on page 4, line 160 shows the meaning.
Response: We thank the reviewer for highlighting these two points for revision and they have both been amended in the text.

Reviewer 2 Report
First of all, I would like to thank the editors the opportunity to review this interesting manuscript entitled “Diet, physical activity, and adiposity as determinants of circulating amino acid levels in a multiethnic Asian population”. After doing a thorough review of the work, in my opinion is necessary to improve some aspects:
Firstly, I would like to thank the authors for their work and implication given the novelty of the study. However, the study has important limitations that need to be reviewed.
The introduction does not provide a sufficient theoretical framework to support the objective of the study. The relationship between lifestyle and dietary protein is discussed. However, a greater theoretical framework is needed to explain the mechanisms related to lifestyle. The authors take into account that lifestyle is the sum of several behaviors, among which we can include physical activity, sedentary time and good nutrition. However, the importance of these behaviors is not introduced in the manuscript.
I suggest that the authors rewrite the introduction, highlighting the importance of lifestyle as well as related behaviors (e.g., body composition, physical activity, diet) and their relationship to amino acid levels. That is, the authors should describe why they think lifestyle factors may be related to serum concentration of amino acids
Methods:
The section of the method is presented in a confusing and disorganized way. I recommend the authors to order this section by sub-sections: design and participants, measure, procedure and statistical analysis
The authors provide an important description of the study sample, however, the number of participants by gender and age remain to be indicated. In addition, it would be interesting to report the mean age and standard deviation for each sex.
The authors said: "Reasonable correlations were found for total energy (r = 0.56) and key nutrients (total fat: r = 0.58; saturated fat: r = 0.51; polyunsaturated fat: r = 0.39; monounsaturated fat: r = 0.50; cholesterol: r = 0.51; protein: r = 0.46). Food and nutrient intakes and total energy intake for each study participant were derived from portion sizes and frequency of consumption and standardized to grams per day. We calculated protein intake as a percentage of total energy. In addition, we calculated intakes of major food sources of protein (red meat, poultry, seafood including fish and shellfish, and soy) based on their relative composition in recipes and mixed dishes."
This information should not be in the Method section. In this paragraph the authors are describing the results of the research. Therefore, please enter this information in the results section.
What international regulations have the authors followed for this research? Please, this information is important to be included.
It would be important for the authors to include two subsections. On the one hand, it is important that the authors describe the procedure carried out during the evaluations. On the other hand, a data analysis section where the statistical procedure followed is explained in detail.
Discussion:
The discussion is greatly enhanced by the description in the introduction. However, it would be interesting if the authors showed a personal explanation of the results of their study. The authors should give an explanation that helps to understand the results found. In addition, I suggest that the authors add a short description of the strengths of their study. For example, the large sample evaluated could be a strength of their work.
Author Response
Reviewer 2
First of all, I would like to thank the editors the opportunity to review this interesting manuscript entitled “Diet, physical activity, and adiposity as determinants of circulating amino acid levels in a multiethnic Asian population”. After doing a thorough review of the work, in my opinion is necessary to improve some aspects:
Firstly, I would like to thank the authors for their work and implication given the novelty of the study. However, the study has important limitations that need to be reviewed.
The introduction does not provide a sufficient theoretical framework to support the objective of the study. The relationship between lifestyle and dietary protein is discussed. However, a greater theoretical framework is needed to explain the mechanisms related to lifestyle. The authors take into account that lifestyle is the sum of several behaviors, among which we can include physical activity, sedentary time and good nutrition. However, the importance of these behaviors is not introduced in the manuscript.
I suggest that the authors rewrite the introduction, highlighting the importance of lifestyle as well as related behaviors (e.g., body composition, physical activity, diet) and their relationship to amino acid levels. That is, the authors should describe why they think lifestyle factors may be related to serum concentration of amino acids
Response: We thank the reviewer for suggesting these improvements to our Introduction section, and hope that the restructured Introduction is satisfactory. The new Introduction now begins by discussing the reasoning behind our focus on modifiable behaviors and mentioning some potential mechanisms responsible for the association between certain lifestyle factors and disease risk (page 1, lines 35-45). We then describe the reasoning behind our use of metabolomics and our focus on amino acids, in that studying these metabolites can address existing knowledge gaps (page 2, lines 46-50).
Methods:
The section of the method is presented in a confusing and disorganized way. I recommend the authors to order this section by sub-sections: design and participants, measure, procedure and statistical analysis
The authors provide an important description of the study sample, however, the number of participants by gender and age remain to be indicated. In addition, it would be interesting to report the mean age and standard deviation for each sex.
The authors said: "Reasonable correlations were found for total energy (r = 0.56) and key nutrients (total fat: r = 0.58; saturated fat: r = 0.51; polyunsaturated fat: r = 0.39; monounsaturated fat: r = 0.50; cholesterol: r = 0.51; protein: r = 0.46). Food and nutrient intakes and total energy intake for each study participant were derived from portion sizes and frequency of consumption and standardized to grams per day. We calculated protein intake as a percentage of total energy. In addition, we calculated intakes of major food sources of protein (red meat, poultry, seafood including fish and shellfish, and soy) based on their relative composition in recipes and mixed dishes."
This information should not be in the Method section. In this paragraph the authors are describing the results of the research. Therefore, please enter this information in the results section.
What international regulations have the authors followed for this research? Please, this information is important to be included.
It would be important for the authors to include two subsections. On the one hand, it is important that the authors describe the procedure carried out during the evaluations. On the other hand, a data analysis section where the statistical procedure followed is explained in detail.
Response: We thank the reviewer for suggesting these improvements for the Methods section. In accordance with these suggestions, which were also mentioned by Reviewer 1, we have divided the Methods section into the following subsections: Study Population and Design (page 2, line 64), Assessment of Dietary Intakes (page 3, line 105), Assessment of Physical Activity and Anthropometric Measures (page 3, line 124), Assessment of Serum Amino Acids (page 4, line 145), Assessment of Covariates (page 4, line 167), and Statistical Analysis (page 4, line 175). We previously mentioned the number of females and males in the overall study population (page 5, line 212), but have added the mean age and standard deviation for each sex (page 5, lines 213-215). Also in accordance with comments from Reviewer 1, we have clarified that the validation of the food frequency questionnaire was published previously and not part of the current study; furthermore we added an additional reference for these published results (page 3, lines 113 and 119). Additionally, we have stated that SP2 and the follow-up examinations were conducted in agreement with the Declaration of Helsinki conventions (page 2, lines 83-84).
Discussion:
The discussion is greatly enhanced by the description in the introduction. However, it would be interesting if the authors showed a personal explanation of the results of their study. The authors should give an explanation that helps to understand the results found. In addition, I suggest that the authors add a short description of the strengths of their study. For example, the large sample evaluated could be a strength of their work.
Response: We appreciate the reviewer’s suggestions for improving the Discussion section, and have enhanced our explanation of the practical significance of our results by highlighting specific health outcomes associated with the significant lifestyle factors we identified (page 13, lines 470-472). Furthermore, we added a short discussion of the strengths of our study, including the large number of ethnic Asian participants as well as the large numbers of male and female participants, which allowed us to conduct stratified analyses by sex (page 13, lines 480-483).

Reviewer 3 Report
I've read with attention the paper of Samuel H. Gunther et al. that investigated the associations between lifestyle factors and adiposity, and serum amino acid profiles in an Asian population. Overall, the manuscript and tables are logically organized and well written in English. The methodology applied is correct, the results are reliable and adequately discussed. In the following, I listed my comment to improve the manuscript (MS):
Introduction
- The introduction is well written and the background and aim of the study have been clearly defined. However, the relationship between amino acid profile and human health and diseases is described marginally. Please provide some more information on the relationship between amino acid profile and health implications to highlight the importance of the study.
Materials and Methods
- The study investigated the relationship between the amino acid profile and adiposity. There are several better, readily available, and widely used methods for assessing adiposity (except BMI and WC) e.g. body composition analysis. A careful study of the relationship between body fat and visceral fat content could be an added value of the study.
Discussion
- line 253-255 -> Editorial error. Please, delete the phrase.
- The discussion is properly written and based on other relative studies. However, the authors should more highlight the novelty of the present study and the practical aspect of the obtained results. Please, consider also discussing more broadly the relationship between different dietary protein sources and the serum amino acid profile.
Round 2
Reviewer 1 Report
Thanks to all the authors for the improvement made in their manuscript.
There are some typos and points still to be improved. More precisely:
1) Please report what is a "Q-values". And in line 191 remove "e" from "control"
2) The "new" restyle of the methods is better than the previous version, but you have to move from "statistical analysys" the list of variables collected.
In this section is necessary to find only the statistical methods, tests and which are the index used to describe and summarize the variables.
3) Please report the test used to obtain pvalues in Table 1.
4) Please report, in the update part (pag.5 line 174), the statistical test or method that you implemented to test the "trend effect".
5) Please could you explain better what do you mean with "...but given the high likelihood of residual confounding in this study, we did not report results from univariate models".
6) I'm agree with you with your consideration on "power" concerning different etnicities but why not try to dicothomize Chinese vs Other to Chinese? And maybe could be an idea for the future of this study?
7) I think, starting from your "strong" reply,
"Results for ethnic Chinese, who made up the majority of our study population, were consistent with the results for the overall population."
that this etnicity effect is really important. Please it could be better to underline more this aspect in the discussion. May be also reporting a reference starting from your consideration: "The population was mostly ethnic Chinese (75.1% overall), followed 199 by Malay (15.1%) and Indian (10.8%), which is consistent with the demographics of Singapore at the 200 national level."
8) Always concerning tables please modifiy the beta-values with beta-coefficients if they are coming from regression output.
9) Please revised this sentence "Strengths of our study include the large number of ethnic Asian participants..."
10) "Our study also included large numbers of male and female participants, which allowed us to conduct stratified analyses and test for interaction effects by sex." I don't feel this is a real strenght to report, please revised or removed.
Reviewer 2 Report
Reviewer 2
I would like to thank the authors for the hard review work done. After a careful review, a better quality of work can be seen, so I must congratulate you on that. However, there is a question that I would like to clarify:
Introduction:
The authors say in line 36 - 37 "Obesity, for example, increases the risk of type 2 diabetes, while physical inactivity and sedentary behavior increase diabetes risk independent of obesity [5]"
As is known, obesity is a disease caused by numerous unhealthy habits, which endanger people's health. However, as indicated by the World Health Organization (2010), obesity is a precursor disease of the metabolic syndrome, and as a consequence, it increases the risk of suffering non-communicable diseases such as cardiovascular diseases, type II diabetes and cancer. Therefore, I suggest to the authors that you please rewrite this sentence.
I suggest you consult the following references:
- World Health Organization. Obesity and overweight. 2019. Available from: https://www.who.int/en/news-room/fact-sheets/detail/obesity-and-overweight
- NCD Risk Factor Collaboration (NCD-RisC). Worldwide trends in body-mass index, underweight, overweight, and obesity from 1975 to 2016: a pooled analysis of 2416 population-based measurement studies in 128·9 million children, adolescents, and adults. Lancet. 2017 Dec 16;390(10113):2627–42.
